# Direct Pulp Capping with Advanced Platelet-Rich Fibrin: A Report of Two Cases

**DOI:** 10.3390/medicina59020225

**Published:** 2023-01-26

**Authors:** Janet N. Kirilova, Dimitar Kosturkov

**Affiliations:** Department of Conservative Dentistry, Faculty of Dental Medicine, Medical University, 1000 Sofia, Bulgaria

**Keywords:** direct pulp capping, regenerative endodontic therapy, platelet-rich fibrin, A-PRF+, ozone therapy

## Abstract

This article aims to prove dentin bridge formation in two cases after direct pulp capping in reversible pulpitis using the platelet concentrate A-PRF+ and preservation of the vitality of the dental pulp. The hemostasis process for the pulp wound and cavity disinfection with gaseous ozone was performed under anesthesia. A large A-PRF+ membrane was prepared from blood plasma and applied to the pulp wound. After placing an MTA, the cavity was closed using glass-ionomer cement. Clinical and cone beam computed tomography findings demonstrated the formation of a dentin bridge in both cases. After the definitive restoration was conducted during the sixth month, the teeth from both patients were asymptomatic and had normal electric pulp testing values. *Conclusions*: Via clinical and CBCT examinations, we observed the dentin bridge formation after placing the platelet concentrate A-PRF+ in both cases. The vitality of the dental pulp was preserved. Further research is needed to refine the clinical protocol, recommended period for control examination, clarification of the precise indications of platelet concentrates, etc.

## 1. Introduction

Preserving the vitality of the dental pulp is of primary importance when treating hard dental tissue diseases. Endodontic treatments after dental pulp damage are complex, time-consuming, and expensive, with limited predictability [1]. Therefore, maintaining the integrity and function of the dental pulp is essential, and the development of materials in the last two decades (mineral trioxide aggregate (MTA) cement) has led to new possibilities in this regard (e.g., a new gold standard). In addition, some disadvantages of calcium silicate cement, such as high cost, manipulative qualities, slow hardening, the presence of arsenic, and others, are described [2]. Dental pulp preservation in the case of pulp communication depends on its functional state, communication size, wound type, and bleeding, among other factors [1,3,4]. According to the authors, calcium silicate cement has succeeded in direct pulp capping in 84.6–100% of cases [5,6]. Studies have shown that optimal clinical outcomes can be obtained in pulp communications up to 1 mm [7,8]. Thus, it is logical to raise the question of whether the same clinical outcomes can be obtained in more extensive communications of the dental pulp.

In recent years, new evidence has accumulated for the regenerative capabilities of dental pulp (the involvement of fibrocytes, undifferentiated mesenchymal cells, the presence of stem cells in the dental pulp, and various immune cells) [1,4]. The presence of components in the dentin matrix (DMC) activated in caries disease to form tertiary dentin has been demonstrated [3].

Rapid advances in biotechnology over the last two decades have allowed for the development of various platelet concentrates and a new approach for pulp therapy: regenerative endodontic therapy with direct pulp capping [1,2,7]. Centrifuging blood separates plasma from the erythrocyte mass. Different centrifuging methods create several generations of platelet concentrates, which can regenerate affected tissues [9,10].

In the literature, cases of preserving the vitality of the dental pulp when treating reversible pulpitis with platelet concentrates are rare [11]. Shobana et al., in a randomized study, correlated results from direct pulp capping with MTA and platelet concentrates [12]. As early as 1995, Rutherford et al. pointed to using growth factors for direct pulp cupping [2].

Various plasma concentrates have used A-PRF+ [12], but no cases have been described involving the administration of advanced platelet-rich fibrin concentrate plus (A-PRF+).

According to Bjørndal et al., there are critical questions about the superiority of one technique over another and the best biomaterial for closing pulp communication when treating reversible pulpitis. There is also the question of whether the open dental pulp is a negative prognostic factor for treatment. More evidence and strategies for managing the exposed dental pulp will be needed in the future [7,8].

Therefore, this article aims to prove dentin bridge formation in two cases after direct pulp cupping in reversible pulpitis using platelet concentrate A-PRF+.

## 2. Methods

**Case 1.** A 47-year-old woman was diagnosed with reversible pulpitis observed in tooth 36 (electropulp test [EPT]: 11 µA, Scorpion, Optica Laser, Sofia, Bulgaria; saturation 82%; Contec CMS 8000D, Qinhuangdao, China). The carious lesion was located on the occlusal surface of the tooth. During the excavation of caries, the lesion was found to communicate with the dental pulp near the mesiolingual pulp corn by approximately 1 mm, with minimal bleeding (Figure 1a). For this treatment, the patient signed a standard informed consent. In addition, due to the use of platelet concentrate, the patient also signed an informed consent to administer A-PRF+. Direct pulp capping was performed using A-PRF+ in accordance with Choukroun’s protocol [9,10].

Under anesthesia, pulp wound hemostasis and cavity disinfection with ozone gas was performed for 24 s using an ozone generator (Prozone, TIP TOP TIPS Sarl, Rolle, Switzerland) [13]. Venous blood (10 mL) was obtained from the patient by venipuncture and centrifuged at 1300 rpm for 14 min (Duo Centrifuge/Process for PRF, Nice, France). An A-PRF+ membrane was made from blood plasma and applied to the pulp wound (Figure 1b). The volume of the inserted membrane was large and filled a significant part of the cavity. The pulp wound was covered by only an A-PRF+ membrane. Above this membrane, MTA cement (Angelus, Londrina, Brazil) was applied to fix the A-PRF+ membrane to the dentin. The calcium silicate cement had no contact with the pulp communication. Therefore, the healing effect on the pulp was from its contact with A-PRF+. The cavity was closed with glass ionomer cement (GIC; Fuji LC II, GC Int. Corp, Tokyo, Japan) (Figure 1c). The patient complained and reported spontaneous pain for several minutes during the next two days without needing medication.

After three months and four days, the treated tooth was asymptomatic. The paraclinical test findings were EPT, 6 µA, and saturation, 84%. Pulse oximetry was performed to observe the blood circulation in the tooth. We used a custom-made holder [14]. The device consists of a monitor and a probe. The probe has two parts—an LED (red and infrared) and a photodetector. The teeth tested are isolated and dried. Then, the LED is placed buccally in the cervical region of the tooth, and the photodetector is placed orally at the same level (Figure 2). On the screen, the saturation level of the pulp is represented as a percentage, indicative of the condition of the pulp. The normal values vary between 82% and 85% [14].

After the resorption of the platelet concentrate, a gap formed since the membrane volume was significant (Figure 1b). Therefore a second visit was required. Under the magnification of an operating microscope, the temporary restoration was carefully removed. A newly formed dental bridge was observed under sterile conditions. The dental pulp was intact (Figure 1d). There was no pain during probing or percussion. On the second visit, there was no observable communication with the pulp. Therefore, we used indirect pulp capping by placing calcium silicate cement—Biodentine (Septodont, Saint-Maur-des-Fossés, France) and a definitive composite restoration (Diamond, Kulzer, Hanau, Germany). In the sixth month after the beginning of the treatment, the tooth was asymptomatic, with an EPT result of 16 µA and a saturation value of 83% [14].

At the sections of CBCT, the formation of a dentin bridge was observed at the site of pulp communication (Figure 1e,f red arrows). The volume of the dental pulp was maintained (the orifices of the mesiolingual and distal root canals were preserved) (Figure 1f). Near the mesiobuccal orifice, the formation of excess reparative dentin was observed. This may attribute to the elevated electric pulp testing values after six months at the tender points in the mesiobuccal cusp. The method for assessing the vitality of the dental pulp comprises measuring its blood circulation. We used a pulse oximeter and dental probes to measure blood circulation. The results obtained showed normal blood circulation (84–83%) from the first until the last day of the examination period, even though the EPT values increased (from 6 to 16 µA) [14].

**Case 2.** A 29-year-old man underwent treatment for reversible pulpitis in tooth 18. A significant carious lesion was present on the buccal surface of the tooth. Communication with the dental pulp at the mesiobuccal pulp horn was observed. The communication was approximately 1.6 mm and exhibited bleeding and mild pain (EPT: 20 μA) (Figure 3a). The active point for the EPT test lies on the part of the enamel that is not supported by dentin. Therefore, the test was performed on palatal cusps, which may have led to values being slightly higher than normal. A saturation examination was impossible due to the sizeable buccal caries (Figure 3). The patient provided additional written informed consent for direct pulp capping with platelet concentrate A-PRF+ according to Choukroun’s methodology [9,10]. The treatment procedure was identical to that in Case 1. The patient complained and reported spontaneous pain for the next two to three days at intervals of several minutes without needing medication.

The patient was not present for the follow-up examination at three months. After five months, the patient visited us after losing part of the temporary restoration. CBCT revealed a dentin bridge at the site of the pulp communication (Figure 3c) and in the suprapulpal dentin adjacent to the defect (Figure 3c). In the CBCT images, a gap is visible above the dentin covering the dental pulp (Figure 3c). A clinical study revealed a lack of communication between the dental pulp and the presence of a thick dentinal bridge (Figure 3b).

Under the magnification of an operating microscope, the remains of the temporary restoration were carefully removed. After placing Biodentine (Biodentine, Septodont, France) and a definitive composite restoration in the sixth month after the beginning of treatment, the tooth was asymptomatic with normal EPT values.

## 3. Discussion

The fibrin membrane, derived from A-PRF+, acts as a scaffold for regenerating the tissue on which it is placed. In this case, the target tissue is the dental pulp (Figure 1c). In addition, this concentrate contains activated growth factors from granules in platelets (activated by blood centrifugation). The essential growth factors for wound healing include the vascular endothelial growth factor, platelet-derived growth factor, transforming growth factor-beta, epidermal growth factor, and insulin-like growth factor [15]. Growth factors stimulate angiogenesis and tissue regeneration during healing. In addition, the autogenic platelet concentrate also contains a significant number of leukocytes, which secrete signaling factors and attract stem cells from the blood [16]. Moreover, stem cells, a well-developed circulatory system, and dormant capillaries are present in the dental pulp [4]. There are three factors for successful tissue engineering—the tissue skeleton, growth factors, and stem cells [17]. The resulting fibrin matrix A-PRF+ possesses porous structures and increased interfibrous spaces (facilitating cell migration); the distribution of platelets is uniform and their numbers increase significantly. Leukocytes are also present in the plasma [10]. PRP and L-PRF products also contain platelets and leukocytes (in smaller amounts). However, there is a high-density fibrin network, thus hindering cell migration [9]. Growth factor levels are significantly higher in A-PRF+ than in A-PRF and L-PRF on the tenth day [10]. Blood cell migration and type 1 collagen synthesis are the highest in A-PRF+. Differences in blood cell distributions are observed between A-PRF and L-PRF. There is an accumulation of neutrophilic granulocytes, lymphocytes, T-lymphocytes, and stem cells towards the end of the clot near the red blood cells [18]. More of these cells are found in A-PRF and A-PRF+ than in L-PRF [18]. For direct pulp capping in the described cases, we used part of the obtained membrane A-PRF+ from an area immediately next to the erythrocyte mass (buffy coat). This is probably the reason for the observed rapid formation of the dentin bridge three months after the treatment in Case 1 and after five months in Case 2.

Using platelet concentrate, the dentin wound is covered with a dentin bridge observed by clinical examination. After three months, when the temporary filling is removed, there is no communication with the pulp in case 1 (after five months in case two). On the second visit, treatment is done on the methodology of indirect pulp capping. This is necessary because there is no communication in cases 1 and 2. Since there is no pulp wound and indirect pulp capping is needed, it is necessary to apply a pulp capping agent [2]. As such, Duncan and co-authors recommend calcium silicate cement or glass-ionomer cement. The authors note that they do not have perforations for either material [2]. The exact indications of which calcium silicate cement to use in different cases have not been described. Our studies found that encapsulated variants of glass-ionomer cement (Fuji LC II) have almost the same biological tolerances as calcium-hydroxide cement (Basic L) [19]. This confirms their application as a possible choice. In these cases, we chose Biodentin because the curing time is less than 15 min [7]. Nevertheless, this is not the only calcium-silicate cement or glass-ionomer cement that can be used.

A study conducted by Kirilova et al. using A-PRF + found a rapid healing process in treating chronic apical periodontitis. Recovery of the bone structure was statistically significant and almost complete at the sixth month of treatment [20].

Miron et al. described successful results with direct pulp capping. The authors applied a minimal quantity of another type of platelet concentrate, i-PRF, in combination with calcium-silicate cement [21]. In treating our patients, the platelet concentrate is A-PRF+, and its amount is significant.

In June 2022, Shobana et al. published a randomized study that compared treatments using direct and various pulp capping agents: MTA, PRP, and PRF [10]. They conducted a CBCT study at 12 months and found a thicker dentin bridge in the groups treated with growth factors. The difference, in our cases, is that we prove the formation of a dentin bridge by CBCT examination after the fifth month of treatment. We also found the formation of a dental bridge at three months for case one and five months for case two following a clinical exam.

In the study by Shobana et al., one group was administered a PRP platelet concentrate, known to have bovine thrombin in the tubes, which slows blood coagulation plasma. Bovine thrombin is a foreign protein for the human body. We use A-PRF+ without additional additives in the tubes; thus, no antigen–antibody reaction can develop [10].

Shobana et al. did not indicate which type of Shoukrouns platelet concentrate was used. We use the platelet concentrate A-PRF+ for the direct pulp capping to form a dentin bridge. The authors studied patients aged 18–45, and our patients were 47 and 29 years old. The authors conclude that PRF platelet concentrates are a promising alternative for direct pulp capping compared to the gold standard for this treatment which is calcium silicate cement [12]. This coincides with the results of the clinical cases we presented. Platelet concentrates are made from the patient’s tissues, while calcium-hydroxide cement is a chemically designed product with some disadvantages.

In the study of Shobana et al., a highly resorbable sterile collagen membrane was used as a carrier of the platelet concentrate, and we covered the membrane A-PRF+ with MTA [12]. Therefore, there is a difference in the volume of the membranes used. We used a large membrane, while in the study of Shobana et al., the membrane used was very small.

To treat dentin, a 17% solution of EDTA is recommended [8]. We use gaseous ozone to treat the pulp wound and dentin. Gaseous ozone stimulates dentinogenesis [22]. Our studies have proven that 24 s of gaseous ozone application removes cariogenic microbes in the cavity after removing infected dentin; it has a disinfection effect [13]. Ozone administration has been shown to improve metabolism in inflamed tissues, enhance tissue oxygenation, and reduce inflammation. In addition, regenerative processes in the dental pulp are supported. This way, less postoperative pain is observed, and the need for endodontic treatment of the treated teeth is eliminated compared to the traditional method [21]. In both cases, we had no evidence of prior dental pulp inflammation. Direct communication with the dental pulp is a trauma to its tissues. Applying ozone gas for 24 s, in addition to disinfection, is essential to stimulate its regenerative processes. Ozone is more tolerant than other means, such as chlorhexidine solution and sodium hypochlorite. Ozone affects pain and reduces the openings of dentin tubules in the place around the communication [23].

The treatment for preserving dental pulp vitality can be accomplished in one or two clinical stages [7]. Bjorndal and co-authors recommend stepwise excavation in two visits to preserve the vitality of the dental pulp. They require two visits between several months, a pulp capping agent, and a second cavity opening. The type of recommended pulp cupping agents is the same (calcium hydroxide cement is added) [7]. We also administered treatment in two visits to keep the pulp alive in these clinical cases.

Applying A-PRF+ platelet content results show a rapid healing effect for direct pulp capping. At the site where the A-PRF+ membrane was placed, the pulp volume was preserved under the formed dentin bridge, the underlying orifice was maintained, and no additional dentin deposition was observed. The results obtained in this study are, however, limited. Further research is needed to refine the clinical protocol, recommended period for control examination, clarify the precise indications of platelet concentrates, etc.

## 4. Conclusions

Via clinical and CBCT examinations, we observed the formation of dentin bridges in two cases after applying the platelet concentrate A-PRF+. The vitality of the dental pulp was preserved (normal EPT values) for six months. Further research is needed to refine the clinical protocol, recommended period for control examination, clarification of the precise indications of platelet concentrates, etc.

## Figures and Tables

**Figure 1 medicina-59-00225-f001:**
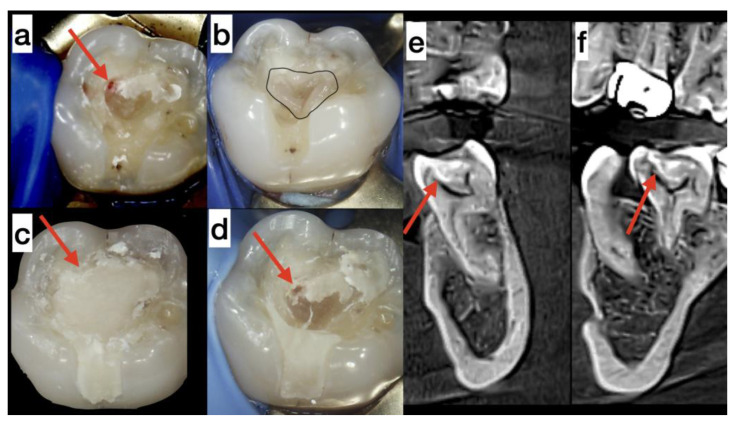
(**a**) Cavity without carious dentin—the red arrow indicates the communication with the dental pulp. (**b**) Application of A-PRF+ membrane on the pulp wound. (**c**) Application of a glass ionomer cement. (**d**) The communication with the pulp was covered by reparative dentin three months later. (**e**) Cone beam computed tomography (CBCT) of tooth 36 after treatment at the coronal section—the red arrow indicates dentin bridge. (**f**) CBCT of tooth 36 after treatment of the sagittal section—the red arrow indicates dentin bridge.

**Figure 2 medicina-59-00225-f002:**
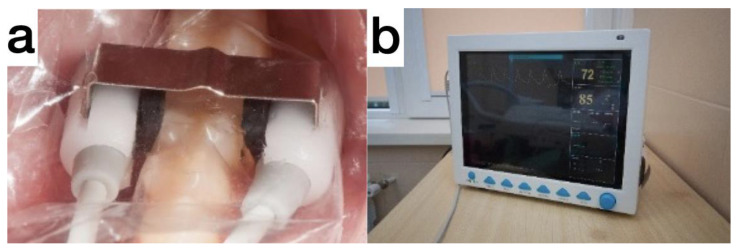
(**a**) The LED is placed buccally in the cervical region of the tooth, and the photodetector is placed orally at the same level. (**b**) On the screen, the saturation level of the pulp is represented as a percentage, indicative of the condition of the pulp.

**Figure 3 medicina-59-00225-f003:**
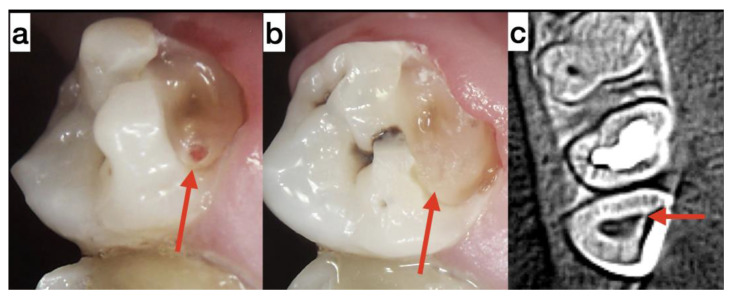
(**a**) Cavity without carious dentin—the red arrow indicates communication with the dental pulp. (**b**) Exposed cavity—dental pulp communication was covered by reparative dentin. (**c**) Cone-beam computed tomography images of tooth 18 five months after treatment (axial section). Dental pulp communication was covered by reparative dentin (red arrow).

## Data Availability

The datasets used and/or analyzed in the current study are available from the corresponding author upon reasonable request.

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
