# Peer review of "Direct Pulp Capping with Advanced Platelet-Rich Fibrin: A Report of Two Cases"

_medicina, 2023, doi:10.3390/medicina59020225_

Round 1
Reviewer 1 Report
The study by Kirilova and Dimitar Kosturkov is about two cases of the successful treatment of direct pulp capping with A-PRF+ and with a follow-up visit (3–5 months). The study can be improved in some cases:
The abstract has not any conclusion section. Please add.
The introduction is so short and did not presents the main gaps.
Discussion is so long and confusing.
The conclusion section is not support the results. It should be re-written
Author Response
Please see the arrachment

Reviewer 2 Report
The authors have described two clinical cases of the treatment of reversible pulpitis using platelet-rich fibrin concentrate plus (A-PRF+).
1. Why did the authors choose this concentrate for the treatment of this nosology? From the introduction, no background on the advantages of this drug for the treatment of complicated forms of caries is formed compared to accepted standard protocols.
2. The protocol of hemostasis and desinfection of the cavity and the point of communication with the tooth cavity also do not belong to the standard category. Why did the authors choose ozone treatment? Has prespontaneous hemostasis been achieved?
3. What is the purpose of applying MTA after applying platelet-rich fibrin concentrate plus (A-PRF +)? Most likely, it was it who contributed to the formation of a dentinal bridge over the communication point - since such an action has long been proven. How can the authors confirm or refute this statement?
4. In the photographs of the second clinical case, there is no isolation of the surgical field using a rubber dam. Is it possible in this case to talk about some kind of unification of treatment protocols?
5. What purpose was Biodentine used in the second stage of treatment?
6. Why are such a small number of cases selected?
The article would be of much greater value if it had a control group and a comparison group, which would really allow us to talk about some advantages of the proposed treatment method.
Reviewer 3 Report
Kindly refer to the attached PDF for all the comments and feedback. Thank you. Best regards

Round 2
Reviewer 2 Report
Dear authors, thank you for the detailed answers to the questions posed!
However, the comments given cast great doubt on the true value/usefulness of using platelet-rich fibrin concentrate plus (A-PRF+) as a direct coating material. It was pointed out that Biodentine was used in both clinical cases "because the formed dental bridge is thin, and the dental pulp can be seen through".
Biodentine was applied at the second visit. Given this, the feasibility of the technique (clinical and financial) looks extremely doubtful, given the need to use a material for classical technique of the direct pulp capping protocol, but on the second visit.
The two clinical cases using platelet rich fibrin concentrate plus (A-PRF+): 1) require an increase in the number of clinical stages and 2) increase the direct and indirect costs of dental clinics. Thus, the obtained clinical results are similar or more likely to be inferior to the known published results of direct pulp capping using materials MTA and Biodentine groups.
In view of the above, these two cases may be of interest in the framework of the internal clinical discussion of specialists, but raise more questions and objections than true usefulness.
Author Response
Thank you for your comments and recommendations. This will improve the quality of the article. We tried to comply with everyone. Below we'll answer all the questions that were asked.
1.However, the comments given cast great doubt on the true value/usefulness of using platelet-rich fibrin concentrate plus (A-PRF+) as a direct coating material. It was pointed out that Biodentine was used in both clinical cases "because the formed dental bridge is thin, and the dental pulp can be seen through.”
Thanks for the note ("because the formed dental bridge is thin, and the dental pulp can be seen through"). Yes, it needs to be changed. Using platelet concentrate, the dentin wound is covered with a dentin bridge established by clinical observation. After three months, when reopening the cavity, there is no communication with the pulp. On the second visit, work is done on the methodology of indirect pulp coating. This is necessary because there is no communication in cases 1 and 2. Since there is no pulp wound and indirect pulp capping is needed, according to the recommendations of the AAE, it is necessary to apply a pulp capping agent[8]. As such, Duncan and co-authors recommend hydraulic calcium silicate cement or glass-ionomer cement. The authors note that they do not prefer this or that material. "Current evidence does not indicate a preference for one material over the other."[2]. The exact indications of which hydraulic calcium silicate cement in which cases to use have not been described. Our studies found that encapsulated variants of glass-ionomer cement (Fuji LC II) have almost the same biological tolerance as calcium-hydroxide cement(Basic L). This does not apply to all types of glass-ionomer cement [ ]. This confirms their application as a possible choice. In these cases, we chose Biodentin because the curing time is less than 15 minutes [ ]. But it could also be another hydraulic calcium-silicate cement or glass-ionomer cement.
2.Biodentine was applied at the second visit. Given this, the feasibility of the technique (clinical and financial) looks extremely doubtful, given the need to use a material for classical technique of the direct pulp capping protocol, but on the second visit.
The application of platelet concentrate is tissue bioengineering. For this reason, a dental bridge is formed, visible in both cases. In the first stage, we close communication with the dental pulp. We conduct an indirect pulp coating on the second visit with a pulp capping agent. Platelet concentrate is a natural alternative for direct pulp capping while using calcium-silicate cement applies chemical products that, although with good qualities, have disadvantages. A-PRF+ is made from the patient's blood. Another author Shobana notes that platelet concentrates are a possible alternative for direct pulp capping.
- The two clinical cases using platelet-rich fibrin contrate plus (A-PRF+): 1) require an increase in the number of clinical stages and 2) increase the direct and indirect costs of dental clinics. Thus, the obtained clinical results are similar or more likely to be inferior to the known published results of direct pulp capping using materials MTA and Biodentine groups.
The data published so far in the literature show better results from the application of platelet concentrates compared to calcium silicate cement. Even after 12 months, Shobana and co-authors proved a thicker dentin bridge. For our two clinical cases, the formation of the dentin bridge is at the 3rd and 5th months of the placement of the A-PRF+ membrane.
About the gap we added to the text:
In case one:
A newly formed dental bridge was observed under sterile conditions. The dental pulp was intact (Figure 1d). There was no pain during probing or percussion. On the second visit there is no observable communication with the pulp. Therefore, we used indirect pulp capping by placing calcium silicate cement -Biodentine (Septodont, France) and a definitive composite restoration (Diamond, Kulzer, Germany).
In discussion:
Using platelet concentrate, the dentin wound is covered with a dentin bridge observed by clinical examination. After three months, when the temporary filling is removed, there is no communication with the pulp in case 1(after five months in case two). On the second visit, treatment is done on the methodology of indirect pulp capping. This is necessary because there is no communication in cases 1 and 2. Since there is no pulp wound and indirect pulp capping is needed, it is necessary to apply a pulp capping agent[2]. As such, Duncan and co-authors recommend calcium silicate cement or glass-ionomer cement. The authors note that they do not have preferations for either material [2]. The exact indications of which calcium silicate cement to use in different cases have not been described. Our studies found that encapsulated variants of glass-ionomer cement (Fuji LC II) have almost the same biological tolerances as calcium-hydroxide cement(Basic L)[19]. This confirms their application as a possible choice. In these cases, we chose Biodentin because the curing time is less than 15 minutes [7]. Nevertheless, this is not the only calcium-silicate cement or glass-ionomer cement that can be used.
One or two clinical stages are possible to preserve the dental pulp's vitality. Bjorndal and co-authors recommend that stepwise excavation in two visits to preserve the vitality of the dental pulp. They require two several-month visits, a pulp capping agent, and a second cavity opening. The type of recommended pulp coating agents is the same(calcium hydroxide cement is added). Yes, dental clinics' direct and indirect costs increase to preserve the vitality of dental pulp, but they reduce the cost of endodontic treatment. And when it comes to complex endodontic anatomy, for example, teeth with taurodontism, C-shaped pulp chamber, middle intermediate root canal, and others, the advantage of preserving the vitality and integrity of the dental pulp the advantages is obvious. And the issue of the indirectly coated pulp coating agent is a matter of choice for the dentist. Additional preparation is needed to implement the methodology, but this is part of the dentist's and his team's complex profession.
4.In view of the above, these two cases may be of interest in the framework of the internal clinical discussion of specialists, but raise more questions and objections than true usefulness.
The development of vital pulp therapy to preserve dental pulp vitality has created considerable interest in this area [Duncan]. However, there needs to be more research on these issues. Future studies are required [Duncan, Bjørndal]. The present paper traces the formation of a dental bridge with a direct pulp coating and ends with an indirect one with the participation of ozone gas. In this way, we respond to the increased interest in the topic in recent years and the accumulated data on the action of platelet concentrates and ozone in medicine. A solution is given to preserve the vitality of the dental pulp. Further developments on these issues, as suggested by Duncan et al., Bjørndal et al., and AAE Position Statement on Vital Pulp Therapy, are necessary.
- Duncan, H.F. Present status and future directions-Vital pulp treatment and pulp preservation strategies. Endod. J. 2022, 55 (Suppl. 3), 497–511.
- Duncan, H.F.; Galler, K.M.; Tomson, P.L.; Simon, S.; El-Karim, I.; Kundzina, R.; Krastl, G.; Dammaschke, T.; Fransson, H.; Markvart, M.; et al. European Society of Endodontology position statement: Management of deep caries and the exposed pulp. Endod. J. 2019, 52, 923–934.
- Bjørndal, L.; Simon, S.; Tomson, P.L.; Duncan, H.F. Management of deep caries and the exposed pulp. Endod. J. 2019, 52, 949–973.
- AAE Position Statement on Vital Pulp Therapy. Endod. 2021, 47, 1340–1344. https://doi.org/10.1016/j.joen.2021.07.015.

Reviewer 3 Report
The manuscript has considerably improved. However, I observed a significant inconsistency in the writing style, which makes some of the explanation difficult to comprehend. Perhaps submitting the work to a proofreading service to improve clarity and grammar will be helpful.
Author Response
Thank you for your comments and recommendations. Style adjustments have
been made, and ambiguities in expressions have been removed.